# Chemosensitization Effect of Seabuckthorn (*Hippophae rhamnoides* L.) Pulp Oil via Autophagy and Senescence in NSCLC Cells

**DOI:** 10.3390/foods11101517

**Published:** 2022-05-23

**Authors:** Uyanga Batbold, Jun-Jen Liu

**Affiliations:** 1Ph.D. Program in Medical Biotechnology, Department of Medical Laboratory Science and Biotechnology, College of Medical Science and Technology, Taipei Medical University, Taipei 11031, Taiwan; uyanga.gemini@yahoo.com; 2School of Medical Laboratory Science and Biotechnology, College of Medical Science and Technology, Taipei Medical University, Taipei 11031, Taiwan; 3Traditional Herbal Medicine Research Center, Taipei Medical University Hospital, Taipei 11031, Taiwan

**Keywords:** seabuckthorn, chemosensitization, autophagy, senescence

## Abstract

The research has demonstrated a synergistic anticancer effect of Seabuckthorn pulp oil (SBO) and the standard chemotherapy regimen Docetaxel (DTX) against two non-small cell lung cancer (NSCLC) cell lines: A549 and H23. The synergizing effect of an SBO and DTX combination was detected utilizing SRB assay and combination index (CI) approaches. Flow cytometry was carried out using fluorescent probes to measure cell cycle analysis by DNA content and reactive oxygen species (ROS) generation. Further, we demonstrated that the synergistic anticancer activity of SBO merged with DTX was achieved by caspase-independent autophagy and senescence induction. These changes were concomitant with increased generation of ROS production and microtubule-associated protein 1 light chain 3 (LC3) protein expression, G1-phase arrest, and enhanced senescence-associated β-galactosidase staining activity. Our data also demonstrated that SBO or DTX treatment groups solely upregulated the phosphorylation of ERK, which coincided with the induction of autophagy vacuoles and was functionally associated with ROS activation. Moreover, endogenous LC3 puncta staining was performed and monitored by confocal microscopy. Overall, these results suggest new mechanisms for the antitumor activity of SBO co-treated with DTX through triggering autophagic cell death and senescence against cancer cells as a result of sustained ERK phosphorylation and intracellular ROS production in NSCLC. In addition, we also highlight SBO as an alternative therapeutic option or adjunct therapeutic strategy in combination with chemotherapeutic agents in lung cancer therapy management.

## 1. Introduction

Lung cancer is the leading cause of cancer-related death worldwide, while the 5-year relative survival rate for all stages of lung cancer is approximately 21%, with the major histological subtype being non-small-cell lung cancer (NSCLC) [1]. The gold standard for NSCLC treatment remains the cisplatin-based chemotherapy regimen. However, prolonged use of chemotherapeutic agents might increase toxicities with a nominal survival rate [2]. In comparison with conventional cancer treatment, traditional medicine is considered a poly-pharmacy with broad-spectrum pharmacological effects through multiple ingredients due to influencing multiple targets [3]. In multifactorial diseases, such as cancer, mono-target treatment is considered more challenging, while multi-target therapies combination of natural phytochemicals with conventional chemotherapeutics tend to potentially enhance the treatment efficacy and decrease the dose-dependent toxicity of the drugs [4].

Seabuckthorn (*Hippophae rhamnoides* L.) is a multipurpose, hardy, deciduous shrub with yellow or orange berries and belongs to the Elaeagnaceae family. The Seabuckthorn is naturally distributed in Asia and Europe and has been used in traditional medicine for centuries for its high nutritional and medicinal properties [5]. This fruit contains naturally occurring antioxidants, including ascorbic acid, tocopherols, carotenoids, and flavonoids, as well as proteins, vitamins, minerals, lipids (primarily unsaturated fatty acids), sugars, organic acids, and phytosterols (Table 1) [6,7]. Different parts of Seabuckthorn, particularly the berries, have been mainly used in traditional medicine practice in Tibet, Mongolia, China, and Middle Asian countries [8]. Studies on animals and humans suggest that Seabuckthorn juice, extracts, oils, seed, and leaves may elicit beneficiary effects on health, including cardioprotective, hepatoprotective, anti-atherogenic, antioxidant, anticancer, immunomodulatory, antibacterial, antiviral, wound healing, and anti-inflammatory properties (Table 1) [9,10,11,12,13,14,15,16]. Seabuckthorn extracts previously showed anti-proliferation activity against malignancies, but the underlying mechanisms of Seabuckthorn pulp oil (SBO) combined with conventional chemotherapeutic agent Docetaxel (DTX) against NSCLC cells still remain unclear.

Docetaxel is a semi-synthetic taxane that possesses almost two-fold higher binding affinity at the β-subunit of tubulin sites than paclitaxel, destabilizing it and eventually leading tumor cells to undergo apoptosis [17]. DTX is considered a standard first- and second-line treatment for patients with NSCLC, as both chemotherapy-naive and platinum-containing chemotherapy-resistant patients show a significant response to the drug [18]. Overall response rates achieved with Docetaxel ranged from 26% to 54% in chemotherapy-naive and 7% to 10% in drug-resistant patients with advanced NSCLC [19].

The mitogen-activated protein kinase (MAPK) signaling pathways, including extracellular signal-regulated kinases (ERK1/2), which are linked to cell proliferation and survival, was found to be activated at a suboptimal dose of taxol. In contrast, more prolonged exposure to a drug or higher concentration resulted in the abrogation of ERK1/2 phosphorylation [20]. It appears that taxol could inhibit ERK or stimulate its expression; this might reactivate tumor progression and thus compromise the efficacy of chemotherapeutic drugs [21,22]. The former study results indicated that ERK activation requires ROS production to initiate and sustain ERK activation to induce cell death [23], which was constant with our experimental results. Activation of ERK1/2 could regulate cell death by apoptosis, while sufficient overexpression of ERK1/2 has also been implicated in autophagy. Although autophagy is considered a cancer cell survival mechanism responsive to cellular stress, recently, extensive autophagy has also led to cell death, pointing to the dual role of autophagy in carcinogenesis [24]. Autophagy can also act as a pro-senescent mechanism, the impairment of which can lead to senescence, though the factors that determine whether cells undergo apoptosis or senescence are not clearly known [25]. Autophagy is the most activated in the G1, S phase limiting the cell growth during stress conditions, while it is inhibited in the G2/M phase.

This study aimed to shed light on the mechanisms of actions of SBO as a neoadjuvant merged with a sub-optimal dose of DTX responsible for the synergizing effects concerning autophagy and senescence, two main events that exert therapeutic properties through the promotion of cell death to control cancer cell survival.

**Table 1 foods-11-01517-t001:** Major-bioactive compounds in Seabuckthorn and their properties.

	Seabuckthorn Active Compounds	Medicinal Properties	References
1.	Tocopherols	Lipid-soluble antioxidant, inhibits oxidative stress, promising diet for Alzheimer’s disease (AD) prevention, may lower cholesterol levels	[7,26]
2.	Carotenoids	Beneficial antioxidant, wound healing through stimulation of angiogenesis, collagen synthesis, epithelialization	[27]
3.	Vitamin C	Antioxidant, acts as an enzymatic cofactor and maintains tissue integrity, and accelerates the formation of skin, epithelial, and endothelial barriers and collagen synthesis	[28,29]
4.	Vitamin B complex (B_1_, B_2_, and B_6_)	Increases the regeneration of axons and neuronal survival, promotes cell repair	[16]
5.	Polyphenolic compounds	Antioxidant, cytoprotective, hepatoprotective, wound healing	[13,14]
6.	Phytosterols	Stimulates microcirculation in the skin, exerts anti-ulcer, anti-atherogenic, and antitumor effects, modulation of the inflammatory process	[11]
7.	Polyunsaturated fatty acids (PUFA)	Immunomodulatory, neuroprotective, anticancer	[30]
8.	Zinc	Capable of improving the blood circulation, anticancer effect, cofactor for a number of enzymes, enhances the utilization of vitamin A	[31]

## 2. Materials and Methods

### 2.1. Antibodies and Reagents

Collection of plant material: Seabuckthorn, a member of the family Elaegnaaceae, (Provisionally accepted name: *Hippophae rhamnoides* L., synonym Elaeagnus rhamnoides (L.) A.Nelson)) is a hardy shrub that is widely found throughout the temperate zones of Asia and Europe. The fruit name has been checked in http://www.theplantlist.org (accessed on 20 November 2020) and http://www.catalogueoflife.org (accessed on 20 November 2020) databases, where the plant name is an accepted name in the genus *Hippophae* L. with a two-star. The record derives from the World Checklist of Selected Plant Families (WCSP) in review, which reports it with original publication details: with the numerical identifier for a plant name in the International Plant Names Index (IPNI)-urn:lsid:ipni.org: names:323851-1. Seabuckthorn fruits were collected from home regions of wild berries in Uvs Province, Mongolia, in the month of October 2020. It is neither endangered nor protected by any legal frame and it is easily and commercially available in the market. The berries were directly frozen for the extraction process.

Chemical reagents including sulforhodamine B (SRB), dichlorofluorescein diacetate (DCFH-DA), propidium iodide (PI), ribonuclease A (RNase A), and trichloroacetic acid (TCA) were from Sigma-Aldrich (St. Louis, MO, USA), while the bicinchoninic acid (BCA) assay kit was from Bio-Rad (Irvine, CA, USA). Antibodies including anti-cyclin E, CDK4, goat anti-mouse immunoglobulin G (IgG)-horseradish peroxidase (HRP), and donkey anti-rabbit IgG-HRP were obtained from Gentex (Irvine, CA, USA), while antibodies against β-actin and MAP-LC3β were purchased from Abcam (Cambridge, UK). Antibodies against caspase-3 and cyclin D were from Cell Signaling Technology (Danvers, MA, USA).

### 2.2. Preparation of Sea Buckthorn Pulp Oil

Firstly, the seabuckthorn seeds and pulp flakes were separated and then stored at −20 °C until they were used in further extraction. The pulp oil was extracted from Seabuckthorn pulp flakes by the solvent extraction method using Dichloromethane (DCM) in a 1:1 (*w*:*v*) ratio. The extraction was carried out at an ambient temperature in the dark with sonication for 12–18 h. After the supernatants were decanted and filtered through filter paper, the DCM solvent of the filtrate was removed using a rotary evaporator at room temperature. Finally, the concentrated extracts were dissolved in ethanol to prepare the stock solution and stored at −20 °C until further use.

### 2.3. Cell Culture

The human A549 (ATCC number: CCL-185) and NCI-H23 (ATCC number: CRL-5800) cells were originally purchased from the American Type Culture Collection (Manassas, VA, USA) and cultured in Dulbecco’s modified Eagle medium (DMEM)/F12 and RPMI 1640 supplemented with 10% fetal bovine serum (Fetal Bovine Serum, Biological Industries, USA), 1% L-Glutamine (Corning, NY, USA) and 100 U/mL (Corning, NY, USA) of 1% penicillin-streptomycin solution. Cells were maintained at 37 °C in a 5% CO_2_ humidified incubator before use in the experiments.

### 2.4. Cell Cytotoxicity Assay

Antiproliferative effects of SBO alone or merged with DTX were measured by the SRB assay, adapted from Houghton [32]. Briefly, A549 and H23 cells were plated in a 96-well plate at a concentration of 3000 cells per well. The cells were pre-incubated overnight and exposed to the SBO at concentrations of 0.1, 0.2, 0.4, 0.8, and 1 mg/mL and/or DTX at concentration of 0.1, 0.5, and 1 nM for a period of 24, 48, and 72 h. At the end of the exposure time, cells were washed with 200 μL PBS twice. Further, the cells were fixed in situ with 50 μL of 10% (*w*/*v*) cold trichloroacetic acid and stained with 0.4% (*w*/*v*) sulforhodamine B (SRB) in 1% acetic acid solution. Finally, 150 μL of 10 mM Tris base solution was added to each well and the plate was shaken on an orbital shaker for 10 min to solubilize the protein-bound dye. The absorbance of each well was read using the SpektraMax iD3 Multi-mode microplate reader (Molecular Devices, Silicon Valley, CA, USA) at 540-nm wavelength, and the cell viability percentage was calculated.

### 2.5. Evaluation of Cell Morphology

Liu’s stain method was performed to observe cell morphology: an 18 mm slide was placed in a 6-well-plate prior to A549 and H23 cells being seeded at 2.5 × 10^4^ cells/well. The next day cells were treated with 1 mg/mL SBO for 24 h, 48 h, and 72 h, and samples were collected. The cells were washed with PBS and Liu A stain was added, followed by Liu B stain. Then slides were gently washed with distilled water, air-dried, and the cell morphology was observed by a light microscope (Leica, Wetzlar, Germany). Photographs were taken with a digital camera (Leica, Wetzlar, Germany).

### 2.6. Drug Synergy

The feasibility evaluation of SBO and DTX was carried out by combining the different doses of SBO (0.5 and 1 mg/mL) and DTX (0.1, 0.5, and 1 nM) exposed for a period of 24 h. The combination index (CI) and the synergistic effect of SBO and DTX were investigated by the Chou-Talalay Method [33]. A synergizing effect occurs when the CI value is <1; additive happens when the CI value equals 1, and antagonism occurs when the CI value is >1.

### 2.7. Cell Cycle Analysis

The cell cycle distribution based on DNA contents was determined by a flow cytometric analysis. A549 and H23 cells were seeded in 6-well plates at a 5 × 10^5^ cells/well density. On the following day, the cells were incubated with SBO (1 mg/mL) and/or DTX (0.1 nM) for 24 h. At test time, the cells were trypsinized, followed by centrifugation at 2000 rpm for 5 min. Subsequently, cold 70% ethanol was added to cells for resuspension. Finally, 3 μL of propidium iodide (PI) stain solution (1 mg/mL) and 5 μL RNase A (1 mg/mL) were added to the samples and then placed for 30 min at room temperature in the dark. The samples were then analyzed on an Attune NxT flow cytometer (Invitrogen, Carlsbad, CA, USA). Data were acquired from 10,000 cells and the percentage of cells in G0/G1, S, and G2/M phases were analyzed using FlowJo software (TreeStar, Ashland, OR, USA).

### 2.8. Intracellular Reactive Oxygen Species (ROS) Level Detection

The production of intracellular ROS was measured by flow cytometry analysis using the DCFH-DA method. Briefly, A549 and H23 cells were seeded in six-well plates at a density of 5 × 10^5^ cells/well and cultured overnight. Then cells were treated with SBO (1 mg/mL) and DTX (0.1 nM) for 24 h. At the end of the exposure time, cells were washed with PBS, collected, and incubated with 5 µM DCFH-DA for 30 min in an incubator. The fluorescence intensity of cells was analyzed on an Attune NxT flow cytometer (Invitrogen, Carlsbad, CA, USA). Data analysis was performed using FlowJo software (TreeStar, Ashland, OR, USA).

### 2.9. Senescence-Associated-β-Galactosidase Staining

The reliable marker for senescent cell detection is the senescence-associated β-galactosidase activity (SA-β-gal) staining assay, which was performed as previously described [34]. Briefly, A549 and H23 cells were plated in 24-well plates at a density of 15,000/well and were incubated in a fresh medium for 24 h prior to adding SBO (1 mg/mL) and/or DTX (0.1 nM) for an additional 72 h. After the exposure time, the cells were fixed in 2% paraformaldehyde, then washed with PBS, and SA-β-gal staining solution was added (1 mg/mL, 5-Bromo-4-chloro-3-indolyl-b-D-galactopyranoside (X-gal)) and incubated for 4 h and examined and photographed under a Leica light microscope (Leica, Wetzlar, Germany).

### 2.10. Immunocytochemistry

A549 and H23 cells were seeded on coverslips in 24-well plates at a density of 15,000/well. After appropriate treatment, the cells were fixated with 4% paraformaldehyde for 10 min at room temperature. Then, they were washed with PBS, and cells permeabilized with 0.1% Triton-X 100 for 5 min. After washing with PBS three times, a blocking solution (5% BSA in PBS) was added and the cells were incubated for 30 min. Upon incubation, the cells were then probed with primary antibody for 2 h at room temperature. The cells were washed with PBS and then incubated with secondary antibody for 1 h at room temperature in the dark. After washing, the cells were stained with Hoechst 33342 for 30 min at room temperature. Finally, the coverslips were mounted on a slide with mounting media and analyzed with a fluorescence microscope (Leica, Wetzlar, Germany).

### 2.11. Western Blot

Briefly, A549 and H23 cells were cultured in 60mm dishes and incubated overnight. Cells were then treated with SBO (1 mg/mL) and/or DTX (0.1 nM) for 24 h. The cells were collected and washed twice with ice-cold PBS and collected in lysis buffer upon exposure time. A bicinchoninic acid (BCA) assay kit was used to determine the protein concentrations in the cellular lysates. An equal amount of proteins of each sample (30 μg) were separated via electrophoresis on sodium dodecyl sulfate (SDS)-polyacrylamide gels and transferred onto nylon membranes for further analysis. The membranes were blocked with 5% non-fat dry milk in TBST (Tris-buffered saline with 0.05% Tween 20) for 1 h and then probed with anti-caspase-3 (diluted 1:1000), anti-pERK1/2 (1:1000), anti-MAP LC3β (1:200), anti-cyclin D (1:1000), anti-cyclin E (1:500), and anti-CDK4 (1:500) at room temperature. Then the membranes were washed with TBS buffer twice following detection with goat anti-rabbit (1:5000) or rabbit anti-mouse (1:10,000) secondary antibodies. After washing the membranes with TBS buffer, immunoreactive bands were detected by the ECL chemiluminescence Western blotting detection kit and exposed to X-ray film (ImageQuant LAS 4000 series) for 1–60 s. β-actin was used as the internal control.

### 2.12. Liquid Chromatographic (LC)-Tandem Mass Spectrometric (MS/MS) Analysis

The analytical LC-MS/MS analysis was carried out on a Waters ACQUITY UPLC I-Class system and Vion IMF QTof MS spectrophotometry, equipped with a Waters BEH C18 Acquity analytical column (75 μm × 150 mm, 1.8 μm). The column oven temperature was set at 40 °C, and the auto-sampler was set at 4 °C. For each run, approximately 2 μL of each sample (1 mg/mL) was loaded onto the column through a 10 μL sample loop using 98% mobile phase A (0.1% formic acid in H2O) at a flow rate of 0.4 mL/min. All experiments were analyzed using positive mode ESI using a LockSpray source. The parameters of LC-HDMSE data were acquired in low energy (MS) and high energy (MSE) mode and were set as follows: mass scan range from *m*/*z* 50 to 1000, capillary voltage 2 kV, the evaporation temperature of 150 °C, and a cone voltage of 30 V.

### 2.13. Statistical Analysis

All statistical analyses were performed using Sigmaplot software (Systat Software, San Jose, CA, USA). Data are represented as means ± standard deviation (SD). Differences between the two groups were calculated using the Student’s t-test and analysis of variance (ANOVA) with pair-wise comparisons. * *p* < 0.05, ** *p* < 0.01, and *** *p* < 0.001 was considered statistically significant.

## 3. Results

### 3.1. SBO Inhibited NSCLC Cells Viability

Initially, the cytotoxicity effect of the Seabuckthorn pulp oil extract (SBO) as a single agent was evaluated by the sulforhodamine B (SRB) assay [35]. As shown in Figure 1a,b, SBO was found to inhibit cell proliferation in the concentration range of 0.1–1 mg/mL, whereas no effect was observed below 0.1 mg/mL in both cell lines. A dose-dependent activity was observed for all time-point experiments in A549 and H23. In contrast, time-dependent effects variation was mainly observed between 24 h and 48 h treatment, but there were no apparent changes between 48 h and 72 h time points. Therefore, the dose of SBO that shows the most potent inhibitory effects (1 mg/mL) was used for further experiments. Although SBO could inhibit cell growth, as a single agent, a comparatively higher concentration of SBO is required to cause a cytotoxic effect in NSCLC.

### 3.2. Synergistic Effect of SBO Combined with DTX in NSCLC

To determine the synergistic effect of SBO and DTX in NSCLC treatment, the combination of SBO and DTX after 24 h of exposure was examined using the SRB assay. A549 and H23 cells were treated with SBO at concentrations of 0.5, 1 mg/mL either alone or merged with DTX at different sub-optimal concentrations of 0.1, 0.5, and 1 nM for 24 h, respectively. As shown in Figure 2, the co-treatment of SBO with DTX markedly reduced cell growth in a dose-dependent manner in A549 and H23 cells. The minimal dose of DTX (0.1 nM) alone caused almost no remarkable cytotoxicity effect in A549 and H23 cells proliferation compared with the control, while the combination treatment with SBO (1 mg/mL) decreased cell viability by 24.6 ± 3.2% and 32.9 ± 4.0%, respectively. Further, the combination index (CI) was used for calculating the synergetic effect, where a CI < 1 indicates synergism, CI = 1 is an additive, and CI > 1 is an antagonism [12]. Table 2 showed that CI values were 0.7 and 0.37 when treated with a minimal dose of DTX (0.1 nM) in combination with 1 mg/mL SBO for A549 and H23 cells, respectively. Therefore, for further investigation of the synergizing mechanism of action of SBO, we used a minimal dose of DTX (0.1 nM), with the reduction in cell proliferation solely limited to up to 5%, and a fixed SBO concentration (1 mg/mL).

### 3.3. SBO Induces Morphological Changes in NSCLC Cells

After a preliminary study of the cytotoxicity of SBO alone or combined with DTX, the morphology changes in A549 and H23 cells were studied. The cells were exposed to SBO at a concentration of 1 mg/mL and/or DTX (0.1 nM), then visualized using Liu’s staining methods. After 24 h of treatment, the cells underwent visible morphological changes. Light microscopy analysis of A549 and H23 cells, particularly in the subpopulation of SBO or combination treatment groups, revealed several morphological changes induced by SBO compared to control cells (Figure 3). Both A549 and H23 cells increased in size and changed to a flattened shape, which are characteristics exhibited in a senescence-like phenotype [36]. Moreover, treated cells also revealed an accumulation of vacuoles in the cytoplasm and a decrease in the cell density relative to control cells (Figure 3).

### 3.4. Synergizing Effect of SBO Induced by G1 Phase Cell Cycle Arrest and Senescence

The cell cycle progression was assessed by flow cytometry with PI staining to analyze whether the cell cycle arrest triggered NSCLC cells inhibition induced by SBO either alone or combined with DTX. The cells were treated with 1 mg/mL of SBO and/or 0.1 nM DTX for 24 h and were collected for cell cycle analysis. The cell population of the G1 phase was increased from 51.3 ± 5.5% and 47.1 ± 1.1% in control cells to 71.8 ± 5.5% and 63.8 ± 5.1% in A549 and H23 cells, respectively, after the indicated treatments (Figure 4). Interestingly, in the combination treatment group with a sub-optimal dose of DTX, cell cycle arrest at the G1 phase was preserved in both cell lines, resulting in an increased G1 cell population to 72.2 ± 8.7% and 58.1 ± 1.5%, respectively in A549 and H23. At the same time, there was a significant decrease in H23 cells in the S phase, whereas in A549 there was a slight alteration in the G2/M phase. The light microscopy observations of Liu’s stain (Figure 3) along with the G1 arrest often implicated in senescence. Senescence is characterized by specific phenotypical and molecular features of the cells along with permanent cell cycle arrest, accompanied by significant morphological alterations [36]. The cells undergoing senescence become flattered and enlarged in size due to the cytoskeleton rearrangement. Senescent cells also have reported an upregulated expression of lysosomal enzyme senescence-associated β-galactosidase (SA-β-gal) during senescence [37]. Thus, we further examined whether the NSCLC cells did indeed undergo senescence. Our results showed that senescence became detectable after treating the cells for 72 h with SBO at indicated concentrations and/or DTX. This proportion of senescent cells dramatically increased in the combination treatment group compared to the control or DTX treatment groups (Figure 4d). Taken together, our results indicate that induction of senescence might contribute to the synergizing effect of SBO merged with a minimal dose of DTX on the proliferation of A549 and H23 cells.

### 3.5. SBO Synergizes DTX Effect via Alteration in Proteins Associated with G1 Arrest and Senescence in NSCLC Cells

Next, Western blot analysis was performed to investigate the molecular mechanisms responsible for SBO-induced G1 phase arrest to examine the levels of major proteins contributing to G1 progression and G1/S transition in the cell cycle. Previously, cyclin D1 activation has been reported to cause a shortened G1 phase, while its downregulation inhibits G1/S transition in many malignancies [38]. The cells were treated with 1 mg/mL of SBO and/or 0.1 nM DTX for 24 h subjected to determine protein expressions associated with cell cycle arrest. As shown in Figure 5a,b, the protein levels of cyclin D, cyclin E, CDK4, and CDK6 were significantly decreased upon treatment with SBO alone or combined with DTX in both cell lines compared to untreated cells. However, no noticeable change was detected in the cyclin E level, specifically in H23 cells. Overall, the results suggested that SBO solely or merged with DTX could effectively trigger cell cycle arrest via downregulating proteins associated with G1 arrest in NSCLC cells.

### 3.6. SBO Induced the Production of Reactive Oxygen Species (ROS) in NSCLC

It has been demonstrated that ROS generation links to entry into senescence and might be activated by many stimuli that also induce autophagy, including oxidative stress [39,40]. Extract of Seabuckthorn leaf previously was reported to induce cell death through induction of ROS generation in prostate cancer and chronic lymphocytic leukemia [41]. Whether SBO induces ROS production in NSCLC cells is not yet investigated. In this study, DFCH-DA staining was used to analyze ROS generation after cell exposure to SBO at a 1 mg/mL concentration and/or DTX 0.1 nM. As shown in Figure 6a,b, there was significant ROS production in A549 and a slight increase in the H23 cell line in both SBO and combination treatment groups. Overall, these results suggest that SBO induces ROS production and, in turn, leads to increased ROS generation in the combination group with DTX, unlike the effect of DTX alone.

### 3.7. Non-Apoptotic Autophagy Is Involved in SBO Induced Synergizing Effect Possibly via Sustained ERK Phosphorylation

To exclude apoptosis as a potent chemosensitization mechanism of SBO, the caspase-3 cleavage pattern was further assessed by Western blot. The A549 and H23 cells were treated with SBO at 1 mg/mL concentration solely or combined with a sub-optimal dose of DTX (0.1 nM) for 24 h. Despite the high level of concentration- and time-dependent synergized cell death observed by SRB assay during the first 24 h of SBO and DTX treatment, no remarkable caspase-3 cleavage was observed in either SBO or combination treatment groups in both NSCLC cells (Figure 7a,b). This result suggests that SBO inhibited cell proliferation when treated alone or with DTX rather than directly killing cells via caspase-dependent apoptosis. ROS generation is reported to be strongly associated with the regulation of autophagy [39]. Figure 6 demonstrates that SBO induced ROS generation in both cell lines when treated solely or combined with DTX. Liu’s stain microscopy observation (Figure 3) of NSCLC cells treated with SBO revealed the formation of cytoplasmic vacuolization that might indicate the cancer cells undergo autophagic turnover. Thus, to validate whether this vacuolization resulted from the induction of autophagy, we performed the Western blot analysis to detect the expression of LC3-II at the protein level, which serves as an ultimate biomarker of autophagy. The data in Figure 7a,b demonstrates that the conversion level of LC3-I to LC3-II was increased after 1 mg/mL of SBO treatment alone or when co-treated with 0.1 nM DTX for 24 h. Further, immunohistochemistry analysis was carried out to confirm the results of the Western blot analysis. Consequently, an increase of LC3 staining and the observation of LC3 puncta, reminiscent of autophagosome formation, were detected in A549 and H23 cells after 24 h of treatment with SBO alone or combination treatment groups relative to the control samples (Figure 7e,f). Recently, evidence indicated a link between ROS generation and autophagy with subsequent activation of cellular signaling molecules, such as mitogen-activated protein kinase (MAPK) [42,43]. Toward this, we examined the effect of SBO alone or merged with DTX on the MAPK signaling pathway. The sustained phosphorylation of ERK has been reported to induce a cell death associated with cytoplasmic vacuolization, suggesting a form of caspase-independent autophagic cell death [44]. ERK plays a vital role in modulating autophagy, we thus have confirmed the increased phosphorylation of ERK by treating SBO either solely or combined with DTX (Figure 7c,d). Since ERK-specific phosphatases are sensitive to ROS, perhaps the sustained ERK activation in NSCLC cells is primarily triggered by ROS activation upon treatment with SBO and/or DTX. Collectively, the above results provided evidence that prolonged ERK activation possibly promotes the death of NSCLC when treated by SBO alone or merged with DTX via inducing autophagy and senescence.

### 3.8. Chemical Profile of SBO

In the Seabuckthorn pulp, the abundance of phenolic compounds and ascorbic acid affects its biological properties and attracts people’s attention. The amount of different constituents varies with the origin of the cultivation, the climate, size, and maturity of the plant, and the extraction and storage procedure [45]. In this study, fresh aerial parts of the Seabuckthorn were collected in the eastern part of Mongolia (Omnogobi, Gurvansaikhan, Mongolia). In this study, an LC-QToF mass spectrometer was used to analyze the Seabuckthorn pulp oil ethanol extract. We list the ten most abundant constituents identified in the SBO ethanol extract in our study. Additionally, the reference sources that are included in the last row indicate several compounds that have been reported in former studies with the potential to induce cell cycle arrest, autophagy, drug synergizing, reverse multidrug resistance and antitumor activities. In our study, deoxycholic acid and lobeline were identified as the dominant constituents in the SBO ethanol extract, followed by poricoic acid A, 24-dehydrocholesterol, and blestrianol B (Table 3). In addition to these findings, the amounts of sterols, flavonoids, phenolic compounds, carotenoids, unsaturated fatty acids, glycoside, and terpenoid contents were evaluated in the whole plant crude extract. For the first time, this investigation reports identifying new triterpenoids, such as iridobelamal A and lupenone, from the extract of Seabuckthorn using the LC-QToF technique.

## 4. Discussion

Natural products are an excellent unexplored source for novel antitumor synergizing agents due to several factors, including unmet therapeutic needs in NSCLC, medicinal plants’ diversity, and low toxicity risk. Previously Seabuckthorn berry was reported for its bioactive compounds with anticancer properties (Table 1). Various Seabuckthorn extracts were compared for their antiproliferative and apoptotic effects on human colon cancer (Caco-2) and human liver cancer cells (Hep G2) [10]. However, its synergizing potential combined with the conventional chemotherapeutic agent DTX remains undiscovered. Here, for the first time, we reported the antiproliferative and synergizing properties of Seabuckthorn pulp oil ethanol extract (SBO) against two NSCLC cells and possible mechanisms of action. We observed that SBO inhibited A549 and H23 cell viability when treated alone in a dose- and time-dependent manner (especially from 24 h to 48 h time points) at comparably higher concentrations than those medicinal herbs reported to possess antitumor properties as a single agent. However, when SBO is combined with DTX at a minimal dose, which alone caused almost no remarkable cytotoxicity, the two-drug combination demonstrates CI indexes equivalent to moderate synergy to synergy effects. However, these antiproliferative and synergizing activities of SBO were not correlated with the direct kill of cells and did not involve a significant activation of caspase-3. Thus, further studies aimed to evaluate whether inhibition of cancer cell proliferation is due to irreversible cell-cycle arrest. The microscopic analysis of Liu’s stain showed the morphological changes, such as cell enlargement, flattening, and the presence of abundant cytoplasmic vacuoles in cells treated with SBO and/or DTX for 24 h relative to control cells, which more possibly indicates the senescence and autophagy. Enlarged cell size, a flattened shape, vacuolization, resistance to apoptosis, cyclin D1 depletion, and senescence-associated β galactosidase (SA-β-gal) activity are all hallmarks of senescence, which was constant with our data [37]. SBO induced G1 phase arrest at cell cycle analysis and activated cellular senescence indicator biomarker expression of Senescence-associated-β-Galactosidase (SA-β-Gal). Here, the SA- β-gal assay showed that SBO induced A549 and H23 cell senescence even more in combined groups than in single-agent groups. Similar to our observation, Boivin D. et al. used the juices of thirteen different berries and revealed that the inhibition of cancer cell proliferation by berry juices did not involve caspase-dependent apoptosis but appeared to induce a permanent cell-cycle arrest, as evidenced by decreased expression of cell-cycle-associated proteins, such as cdk4, cdk6, cyclin D1, and cyclin D3 [54]. To our knowledge, this is the first report showing that SBO induces senescence in NSCLC cells either when treated alone or in combination with DTX.

We failed to observe the apparent induction of cell apoptosis after exposure to SBO solely or merged with DTX, suggesting that apoptosis may not play a predominant role in the cytotoxicity of the combination treatment. Instead, our research data further observed that autophagic cell death is required for the cytotoxic effects of the combination group. Although autophagy was initially described as a protection mechanism in cancer cells against various stress inducers and led to the debate about its role in cancer therapy, excessive autophagy could be identified as another mode of cell death [55]. In the present study, Liu’s stain microscopic observation of A549, H23 cells revealed the presence of cytoplasmic vacuolization treated with 1 mg/mL of SBO and/or 0.1 nM of DTX for 24 h. LC3 is another reliable marker of autophagosomes. During autophagy, the cytosolic form of LC3 (LC3-I) is conjugated to phosphatidylethanolamine to form LC3-phosphatidylethanolamine conjugate (LC3-II), which is recruited to autophagosomal membranes [56]. Here, Western blot and immunohistochemistry analysis of LC3 revealed a significant overexpression and formation of LC3 puncta upon treatment with SBO and DTX co-treatment group after 24 h. The MAPK is a downstream signaling pathway of ROS and plays a pivotal role in inducing apoptosis and autophagy. Many studies have suggested that ROS generation might cause cell death through the activation of MAPK [57]. The former research by Wang et al. proposed that although moderate or short-term activated MEK/ERK signaling molecules might inhibit cell autophagy, sustained and long-term activation could induce autophagy [58]. Accordingly, our study demonstrated that SBO exerted the synergizing activity of conventional chemotherapeutic agent DTX is associated with induction of autophagy via sustained activation of ERK that ultimately led to cell death.

In contrast, another study’s results demonstrated that the antitumor activity of Seabuckthorn could be attributed to particularly phenolic compounds such as flavonoids, including kaempferol, quercetin, and isorhamnetin, where isorhamnetin possesses anti-proliferation activity on lung cancer cells in vitro through induction of apoptosis by the down-regulation of oncogenes and up-regulation of apoptotic genes [45]. Moreover, the anticancer effects of distinct extracts of berries, including Seabuckthorn, were studied on the viability of HT29 and MCF-7 cancer cells in correlation with some carotenoid levels and vitamin C levels. Intriguingly, when the ascorbate standard was exposed alone, it did not show cancer cell cytotoxicity in the same way, suggesting a synergistic activity of vitamin C and other components of the extracts [59].

Initially, our project aimed to utilize whole pulp oil extract to show the possible drug synergizing activity in combination treatment regimens. Thus, at this stage, we could only predict the possibility of some active components possessing chemosensitization properties in NSCLC. We highlight two significant constituents, deoxycholic acid and lobeline, identified in SBO by an LC-QToF mass spectrometer analysis. They were reported to induce autophagy and chemosensitization effects previously. The former study showed that when Barrett’s esophagus (BE) cells were exposed to deoxycholic acid (DCA), Beclin-1 expression was increased along with pronounced induction of autophagy observed by electron microscopy and immunohistochemistry analysis of the GFP-LC3-positive puncta pattern [46]. Moreover, a piperidine alkaloid from Lobelia inflata, lobeline showed the potential to reverse multidrug resistance (MDR) and sensitize drug-resistant tumor cells to doxorubicin through inhibiting P-gp activity at non-toxic concentrations [47,48].

It is possible to conclude that Seabuckthorn pulp oil is a promising extract containing many dietary as well as medicinal compounds with a potentially beneficial role in human health. Moreover, the results support that a minimal dose of chemo drug (DTX) can significantly inhibit cancer cell viability when combined with SBO. Although different from traditional Western pharmaceutical research, this study failed to find a single component with an exact active action but only found some active ingredients reported in the literature. However, this study points out that daily consumable functional foods, such as SBO, can possibly provide pharmacological effects. Studies to elucidate the bioactive compounds these functional foods contain are sufficient as a paradigm for the modernization of diet to treat malignant tumors.

## Figures and Tables

**Figure 1 foods-11-01517-f001:**
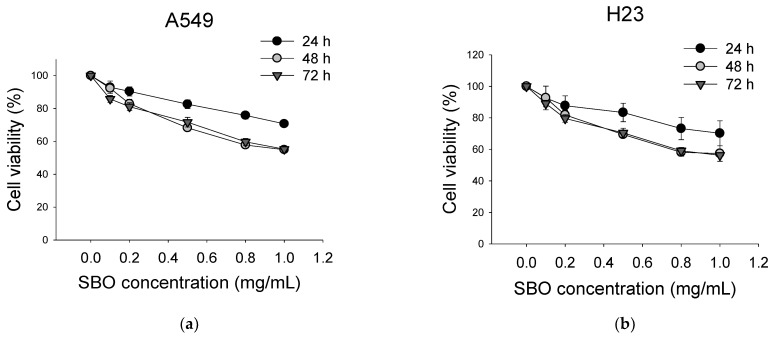
Antiproliferative effects of Seabuckthorn pulp oil (SBO) on the growth of non-small cell lung cancer (NSCLC) cells. The cells were treated with concentrations of 0.1–1 mg/mL of SBO for 24 h, 48 h, and 72 h. Dose-dependent cell cytotoxicity of SBO was determined on the A549 (**a**) and H23 (**b**). Cell viability was evaluated using the sulforhodamine B (SRB) assay. Data are expressed as the mean ± SE.

**Figure 2 foods-11-01517-f002:**
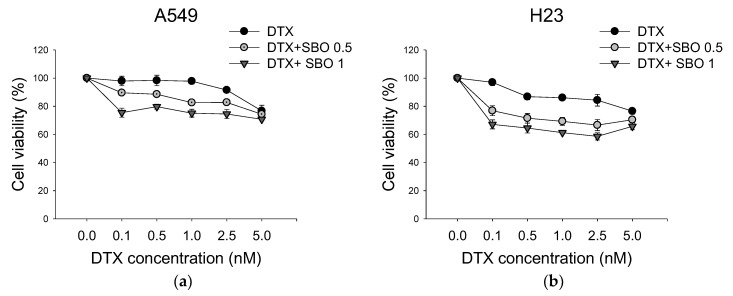
Combinatorial effect of SBO and Docetaxel (DTX) on cell viability of NSCLC. Dose-response curves of (**a**) A549 and (**b**) H23 cells treated with SBO and DTX at indicated concentrations, either alone or in combination, for 24 h.

**Figure 3 foods-11-01517-f003:**
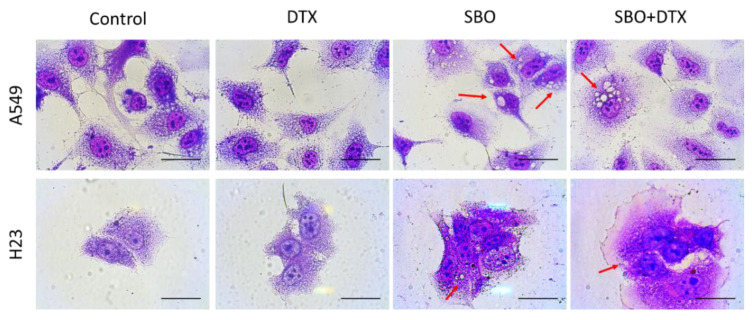
Morphological changes induced by SBO-treated NSCLC cells. Morphological changes in A549 and H23 cells treated with 1 mg/mL SBO solely or co-treated with 0.1 nM DTX for 24 h. The cells were stained with Liu’s stain and visualized under a light microscope. Red arrows indicate cytoplasmic vacuolization. The scale bar represents 50 µm.

**Figure 4 foods-11-01517-f004:**
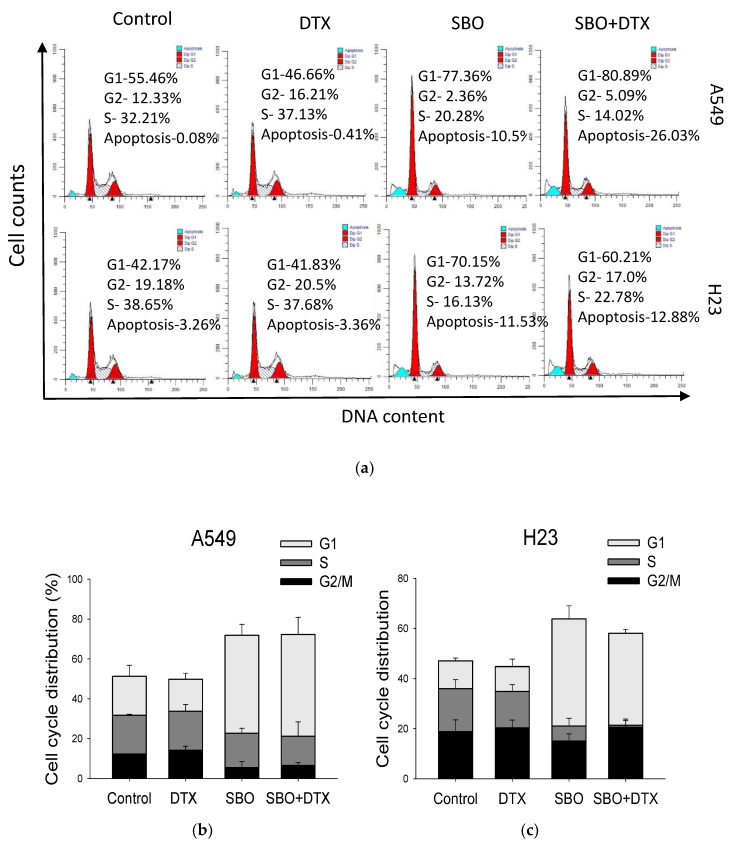
SBO synergizes the DTX effect via cell cycle arrest and senescence in NSCLC. (**a**) The cell cycle distribution treated with SBO (1 mg/mL) solely or co-treated with DTX (0.1 nM) was assessed by flow cytometry analysis after 24 h of exposure. (**b**,**c**) Quantitative data of the percentage of the cell populations in G0/G1, S, and G2/M phases. Data are expressed as mean ± SD. (**d**) A549, H23 cells were seeded in 24-well plates cultured in fresh medium for 72 h and treated with SBO (1 mg/mL) and/or DTX (0.1 nM) for another 72 h. The phase-contrast analysis and SA-β-gal assay were performed. Representative images were photographed under a light microscopic analysis to show SA-β-gal positive cells (blue cells). The scale bar represents 100 µm.

**Figure 5 foods-11-01517-f005:**
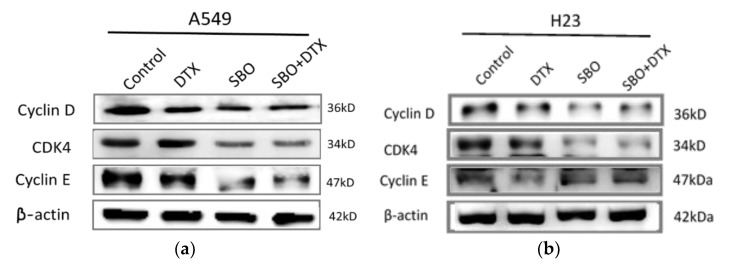
SBO synergizes the DTX effect by inhibiting cell cycle expression and senescence regulators in A549 and H23 cells. (**a**,**b**) After appropriate treatment, the cells were subjected to a Western blot of protein expressions: cyclin D, cyclin E, and CDK4. (**c**,**d**) Densitometry analysis of relative proteins to β-actin was performed and was statistically quantified. Data are the mean ± SD; * *p* < 0.05, ** *p* < 0.01 vs. the corresponding control.

**Figure 6 foods-11-01517-f006:**
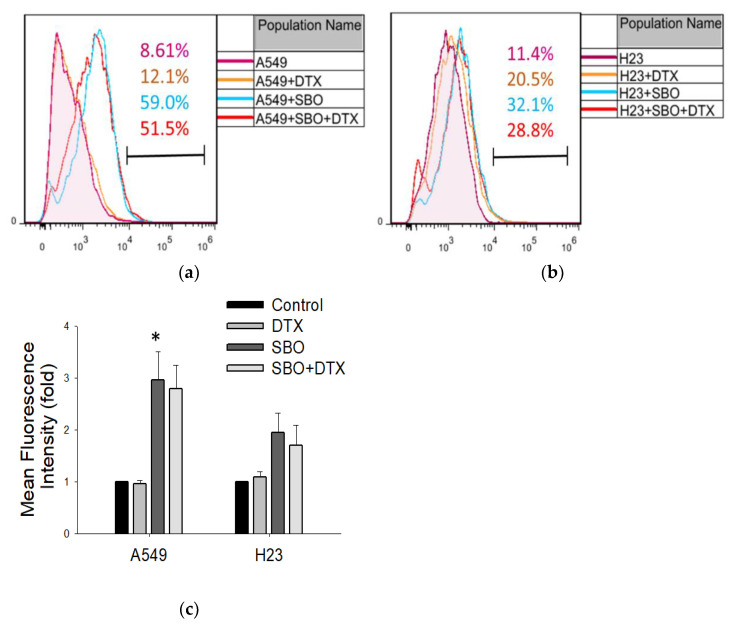
SBO, but not DTX alone, induced the activation of reactive oxygen species (ROS) in NSCLC cells. (**a**,**b**) A549 and H23 cells were treated with SBO at indicated concentrations and/or DTX for 24 h. DCFH-DA assay was performed to determine intracellular ROS level using flow cytometry. (**c**) Histograms represent the statistically quantified ROS production (MFI, mean fluorescence intensity). Data are the mean ± SD; * *p* < 0.05 vs. the corresponding control.

**Figure 7 foods-11-01517-f007:**
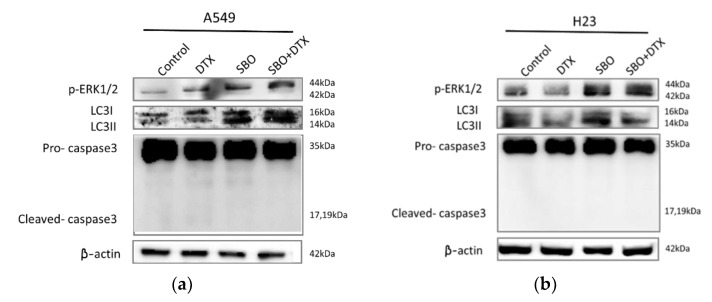
Autophagy induced by SBO is associated with constitutive ERK1/2 activation. (**a**,**b**) A549 and H23 cells treated with 1 mg/mL of SBO and/or DTX 0.1 nM for 24 h and were analyzed by Western blot with antibodies against LC3-I/II, caspase-3, p-ERK1/2. (**c**,**d**) Densitometry analysis of relative proteins to internal control β- actin was performed and statistically quantified. Data are the mean ± SD; * *p* < 0.05, ** *p* < 0.01 vs. the corresponding control. Immunohistochemical staining was performed to analyze the expression level of LC3 in (**e**) A549 and (**f**) H23.

**Table 2 foods-11-01517-t002:** Combination index (CI) values at different levels of growth inhibition induced by SBO merged with DTX.

Cell Line	SBO (mg/mL)	DTX 0.0 nM	DTX 0.1 nM	DTX 0.5 nM	DTX 1 nM
**A549**	0.0		2.1 ± 3.4	1.7 ± 2.9	6.0 ± 6.3
	0.5	10.9 ± 1.5	10.4 ± 1.6 (1.06)	16.2 ± 1.0 (0.64)	17.4 ± 1.2 (0.6)
	1	19.8 ± 2.5	24.6 ± 3.2 (0.7)	22.4 ± 4.6 (0.86)	24.9 ± 2.8 (0.8)
**H23**	0.0		3.0 ± 2.8	9.2 ± 1.6	14.0 ± 2.1
	0.5	22.8 ± 2.7	23.2 ± 4.3 (1.0)	27.7 ± 1.4 (0.6)	30.8 ± 3.6 (0.5)
	1	26.5 ± 7.7	32.9 ± 4.0 (0.37)	35.2 ± 2.9 (0.4)	38.8 ± 1.1 (0.3)

Values were expressed as mean ± SD (*n* = 3). Values in brackets are combination indexes (CI < 1, synergism; CI = 1, additive effect; CI > 1, antagonism).

**Table 3 foods-11-01517-t003:** The main identified constituents of the Seabuckthorn pulp oil ethanol extract (SBO).

No.	Compound Name	Compound Formula	Observed RT * (min)	Observed *m*/*z*	Mass Error (mDa)	References
1.	Deoxycholic acid	C_24_H_40_O_4_	17.46	393.2979	−5.2	[46]
2.	Lobeline alkaloid	C_22_H_27_NO_2_	14.42	338.2116	0.3	[47,48]
3.	Poricoic acid A	C_32_H_48_O_5_	6.57	513.3618	8.4	[49]
4.	24-Dehydrocholesterol	C_27_H_44_O	9.74	385.3494	7.7	[50]
5.	Blestrianol B	C_37_H_32_O_7_	18.72	589.222	−0.2	-
6.	Iridobelamal A	C_30_H_50_O_4_	8.5	475.3809	5.8	[51]
7.	Esculentoside O	C_35_H_54_O_10_	6.37	635.3841	8.1	-
8.	Prosapogenin 1	C_47_H_76_O_18_	18.03	707.3985	−2.2	-
9.	Lupenone	C_30_H_48_O	9.85	425.3797	4.5	[52,53]
10.	Campesterol-β-D-glucoside	C_34_H_58_O_6_	15.62	563.4281	−4.4	-

* RT, retention time.

## Data Availability

The data presented in this study are available on request from the corresponding author. The data are not publicly available due to privacy concerns.

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
