# Peer review of "Chemosensitization Effect of Seabuckthorn (Hippophae rhamnoides L.) Pulp Oil via Autophagy and Senescence in NSCLC Cells"

_foods, 2022, doi:10.3390/foods11101517_

Round 1

Reviewer 1 Report

The study is well-written, and the experimental design correct. The methods are well described, and the results are well interpreted and discussed. The data obtained are well presented. 

I suggest correcting typos along the text to improve the quality of the manuscript.

I also recommend specifying in lines 131-132 why these SBO and DTX concentrations were chosen (add bibliographical references to this if necessary).

Author Response

We sincerely appreciate all valuable comments and suggestions of the Reviewers’ concerning our manuscript entitled “foods-1729474”. We have carefully considered the comments and tried our best to address every one of them accordingly. The main corrections in the paper and our responses to all the comments are as follows:

Comments and Suggestions for Authors

The study is well-written, and the experimental design is correct. The methods are well described, and the results are well interpreted and discussed. The data obtained are well presented.

  1. I suggest correcting typos along the text to improve the quality of the manuscript.

Thank you for the suggestion. We have searched them through the whole article and improved the wording.

  1. I also recommend specifying in lines 131-132 why these SBO and DTX concentrations were chosen (add bibliographical references to this if necessary).

We appreciate your comments. In the dose selection of SBO, taking into account the cytotoxic effect of the solvent (ETOH) used to prepare the SBO stock solution, a maximum concentration of 1 mg/mL is applicable. Therefore, the highest dose of SBO used is 1 mg/mL. As for DTX, the drug has an IC50 value of approximately 2.5 nM in preliminary dose-response studies. However, from our previous findings, suboptimal doses of DTX in combination with other drugs were selected to achieve the highest synergistic effect of the two-drug combination. Therefore, this fact favors the selection of the minimum dose of DTX in the combination regimen.

Reviewer 2 Report

Dear Authors,

The manuscript (foods-1729474) submitted for review is very interesting and it should be made a minor correction by the Authors.

The search for alternative or complementary treatment options is very important, especially for lung cancer, which is the leading cause of cancer death worldwide. In this context, the obtained research results are interesting and promising.

While the manuscript is innovative and interesting, as well as well written, I have a few comments.

Authors, Please note and address the following comments:

Graphical abstract

Although the graphical abstract looks interesting and encourages you to read the article, I would suggest adding an extension for some abbreviations, e.g. Seabuckthorn pulp (SB), and NSCLC (Non-small-cell lung cancer). What did the authors mean by adding G0 / G1 arrest to the graphical abstract?

Introduction: The background of this study is well written.

Material and methods

  • The research methodology is described in detail, but in my opinion, a research scheme would be useful for a better understanding of the experiment.
  • I did not find information about the approval of the bioethics committee for this study, and the human A549 and A23 cells were used in the study.
  • Does point 2.12 concern the methodology of determination of compounds in the Sea buckthorn pulp oil in ethanol extract (SBO? This was not specified in the methodology and should be supplemented.

Limitation

I have a question. Is there any limitation to these results? If yes, it is worth writing about it.

Conclusion

There is no Conclusion section. Perhaps the last two sentences (lines 507-510) would fit the Conclusion. The authors should indicate the directions for further research. What are the authors' recommendations for scientists?

References – I am impressed with the references, 31 of them are from 2010-2022 years, 27 references are from  2000-2009, and 2 references are from before 2000. References are cited according to journal rules.

Technical notes

Table 1 - Have the authors used Standard Deviation in this table? What do the values in brackets in Table 1 in columns 4, 5, and 6  mean? This should be explained below the table.

Figures 1,2,3 - In my opinion, in the headings of these figures, the authors should add the A549 cell and the H23 cell.

Table 2 - The references are used in table 2. Does this mean that other authors have identified the same compounds as the authors of this study? This should be clarified because, with the current record, I had doubts as to whether the authors provided their results or the results of other authors.

I believe this manuscript addresses an important area of research in an international context.

Reviewer

Author Response

We sincerely appreciate all valuable comments and suggestions of the Reviewers’ concerning our manuscript entitled “foods-1729474”. We have carefully considered the comments and tried our best to address every one of them accordingly. The main corrections in the paper and our responses to all the comments are as follows:

Comments and Suggestions for Authors

Dear Authors,

The manuscript (foods-1729474) submitted for review is very interesting and it should be made a minor correction by the Authors.

The search for alternative or complementary treatment options is very important, especially for lung cancer, which is the leading cause of cancer death worldwide. In this context, the obtained research results are interesting and promising.

While the manuscript is innovative and interesting, as well as well written, I have a few comments.

Authors, Please note and address the following comments:

Graphical abstract

  1. Although the graphical abstract looks interesting and encourages you to read the article, I would suggest adding an extension for some abbreviations, e.g. Seabuckthorn pulp (SB), and NSCLC (Non-small-cell lung cancer). What did the authors mean by adding G0 / G1 arrest to the graphical abstract?

Thank you for your kind suggestion. The graphic summary has been corrected according to your request. Here, visualizing it as a graphical summary, we wanted to highlight SBO activity to induce G0/G1 arrest by altering the expression of major proteins that contribute to G1 progression and G1/S transition in the cell cycle thought to be senescence symbols.

Introduction: The background of this study is well written.

Material and methods

  1. The research methodology is described in detail, but in my opinion, a research scheme would be useful for a better understanding of the experiment.

We appreciate your comments. This study is designed to observe the chemosensitization mechanism of SBO using two lung cancer cell lines. For this reason, at this stage, we utilize a visual graphic summary to explain the possible mechanism of action of SBO binding to DTX and to make it easier for the reader to understand by linking the different changes involved in cell death pathways.

  1. I did not find information about the approval of the bioethics committee for this study, and the human A549 and A23 cells were used in the study.

Thank you for your comment. Indeed, the cultivation of cells from tissue requires ethical approval. However, our study did not involve primary cell culture. Instead, human A549 and H23 cells were directly purchased from the cell bank of the American Type Culture Collection (ATCC).

  1. Does point 2.12 concern the methodology of determination of compounds in the Sea buckthorn pulp oil in ethanol extract (SBO? This was not specified in the methodology and should be supplemented.

The answer is yes. Paragraph 2.12 in the method section is the analysis condition used for SBO extraction analysis. Thank you for addressing this point.

Limitation

  1. I have a question. Is there any limitation to these results? If yes, it is worth writing about it.

Thanks for your comment. As a food, this study points out the need for relatively high concentrations of SBO to cause cytotoxic effects in NSCLC. This observation is described in section 3.1. Since the original goal of this study was to use whole fruit pulp extract to demonstrate food may enhance the synergistic effect of chemotherapeutic drugs in NSCLC. Therefore, we solely used alcohol extracts for compound identification instead of using complex separation techniques. For this reason, we cannot determine whether the findings of this study represent a sustainable source of active ingredients in SBO. Therefore, we can only expect the possibility that some active ingredients have chemosensitizing properties. This limitation has been addressed and highlighted in the discussion section. 

Conclusion

  1. There is no Conclusion section. Perhaps the last two sentences (lines 507-510) would fit the Conclusion. The authors should indicate the directions for further research. What are the authors' recommendations for scientists?

We appreciate your opinion. We have highlighted possible directions for further research in the future in the Discussion section of the last paragraph. 

References– I am impressed with the references, 31 of them are from 2010-2022 years, 27 references are from 2000-2009, and 2 references are from before 2000. References are cited according to journal rules.

Thank you for your comment.

Technical notes

  1. Table 1- Have the authors used Standard Deviation in this table? What do the values in brackets in Table 1 in columns 4, 5, and 6 mean? This should be explained below the table.

Thank you for your kind notice. Yes, we have used SD. The correction was made in accordance.

  1. Figures 1,2,3- In my opinion, in the headings of these figures, the authors should add the A549 cell and the H23 cell.

Thank you for your suggestion. To ensure consistency in writing, we would like to keep the initial formatting of the figures.

  1. Table 2 -The references are used in table 2. Does this mean that other authors have identified the same compounds as the authors of this study? This should be clarified because, with the current record, I had doubts as to whether the authors provided their results or the results of other authors.

Thank you for your kind notice. We have listed the reference sources that previously reported potential anticancer effects of the same pure compound we have identified in our study. The correction was done in accordance.
